# Magnetic Carbon Quantum Dots/Iron Oxide Composite Based on Waste Rice Noodle and Iron Oxide Scale: Preparation and Photocatalytic Capability

**DOI:** 10.3390/nano13182506

**Published:** 2023-09-06

**Authors:** Wanying Ying, Qing Liu, Xinyan Jin, Guanzhi Ding, Mengyu Liu, Pengyu Wang, Shuoping Chen

**Affiliations:** College of Materials Science and Engineering, Guilin University of Technology, Guilin 541004, China; 2120210327@glut.edu.cn (W.Y.); 2120220336@glut.edu.cn (Q.L.); 2120200261@glut.edu.cn (X.J.); 1020220195@glut.edu.cn (G.D.); 2120210291@glut.edu.cn (M.L.); 2120220368@glut.edu.cn (P.W.)

**Keywords:** waste rice noodle, iron oxide scale, carbon quantum dots/iron oxide composite, photocatalytic capability, magnetic

## Abstract

To provide an economical magnetic photocatalyst and introduce an innovative approach for efficiently utilizing discarded waste rice noodle (WRN) and iron oxide scale (IOS), we initially converted WRN into carbon quantum dots (CQDs) using a hydrothermal method, simultaneously calcining IOS to obtain iron oxide (FeO_x_). Subsequently, we successfully synthesized a cost-effective, magnetic CQDs/FeO_x_ photocatalytic composite for the first time by combining the resulting CQDs and FeO_x_. Our findings demonstrated that calcining IOS in an air atmosphere enhanced the content of photocatalytically active α-Fe_2_O_3_, while incorporating WRN-based CQDs into FeO_x_ improved the electron-hole pair separation, resulting in increased O_2_ reduction and H_2_O oxidation. Under optimized conditions (IOS calcination temperature: 300 °C; carbon loading: 11 wt%), the CQDs/FeO_x_ composite, utilizing WRN and IOS as its foundation, exhibited exceptional and reusable capabilities in photodegrading methylene blue and tetracycline. Remarkably, for methylene blue, it achieved an impressive degradation rate of 99.30% within 480 min, accompanied by a high degradation rate constant of 5.26 × 10^−3^ min^−1^. This composite demonstrated reusability potential for up to ten photocatalytic cycles without a significant reduction in the degradation efficiency, surpassing the performance of IOS and FeO_x_ without CQDs. Notably, the composite exhibited strong magnetism with a saturation magnetization strength of 34.7 emu/g, which enables efficient and convenient recovery in photocatalytic applications. This characteristic is highly advantageous for the large-scale industrial utilization of photocatalytic water purification.

## 1. Introduction

In recent years, there has been growing interest in using semiconductor materials [1,2,3] as photocatalysts [4,5] for degrading low concentrations of organic and inorganic molecules in freshwater treatment [6], environmental remediation [7,8], industrial [9], and health applications [10,11]. Various types of semiconductors, such as TiO_2_ [12], ZnO [13], ZnS [14], CdS [15], and others, have been reported for photocatalytic decontamination over the past two decades. However, one major challenge encountered with most photocatalysts in powder form is their non-magnetic nature, making them difficult to separate from the purified solution. To address this issue, coupling [16,17,18,19] magnetic particles with non-magnetic semiconductor photocatalysts appears to be the most logical solution. By incorporating magnetic particles into the hybrid photocatalysts, an external magnetic field can be used to easily separate them after the photocatalytic process, ensuring reusability and offering a promising approach for environmental pollution control. Among the various magnetic particles, iron oxides—such as Fe_3_O_4_ [16,20], α-Fe_2_O_3_ [21], β-Fe_2_O_3_ [22], γ-Fe_2_O_3_ [23], FeO [24], and spinel ferrites (MFe_2_O_4_ [25])—have garnered significant attention in the field of photocatalysis due to their low cost, strong magnetism, environmental friendliness, and stability. However, in comparison to classical photocatalysts like TiO_2_ and ZnO, iron oxides do not demonstrate outstanding photocatalytic efficiency. Consequently, it has become essential to explore certain degrees of doping or composite modification to enhance their performance [26,27,28]. Among these, the composition of carbon quantum dots (CQDs) has proven to be an effective means of enhancing the photocatalytic performance of iron oxide materials [29,30,31,32,33,34]. There have been reports demonstrating that carbon quantum dots/iron oxide photocatalytic materials can efficiently degrade water-soluble organic pollutants such as methylene blue. For instance, Sun et al. prepared quantum dots using glucose as a carbon source and combined them with commercially-available magnetic Fe_3_O_4_ nanoparticles to create CQDs/Fe_3_O_4_ composite material. This composite achieved an 83% degradation rate of alkaline methylene blue solution within 30 min [29]. Additionally, Zhang et al. reported the synthesis of CQDs/MIL-101(Fe)/g-C_3_N_4_ composite photocatalytic material, using FeCl_3_ as the iron source and citric acid and ethylenediamine as carbon sources. This composite achieved a photocatalytic rate of 99.3% for methylene blue within 120 min [34]. However, it is worth noting that the CQDs/iron oxide photocatalytic materials reported to date have typically used commercially available iron and carbon sources, resulting in relatively high synthesis costs, which may pose challenges for large-scale commercialization.

The improper disposal of cooking waste has the potential to introduce pollutants into the soil and water ecosystems, posing a considerable urban governance dilemma [35]. The contemporary methods employed for the management of cooking waste in commercial settings [36,37] predominantly encompass techniques like anaerobic digestion [38], aerobic composting [39], landfill deposition [40], incineration [41], and the creation of forage [42]. Despite enabling the large-scale industrial treatment of cooking waste, these strategies are accompanied by notable limitations [43], including substantial land utilization, substantial capital investment in equipment, diminished product profit margins, and the potential for generating secondary pollutants like greenhouse gas emissions and leachates. In order to address these issues, our previous investigation proposed a novel approach to utilizing cooking waste, specifically waste rice noodle (WRN), with starch as the main component. This approach involved first hydrothermal carbonizing the WRN to create a solution of CQDs, which was then combined with specific inorganic nano powders like TiO_2_ or ZnO to produce CQDs/TiO_2_ [44] or CQDs/ZnO [45] photocatalytic composites for water pollution control. The CQDs/inorganic oxide composites derived from WRN displayed remarkable photocatalytic degradation efficiency, outperforming commercial TiO_2_ or ZnO, particularly under visible light illumination, for a range of water-soluble dyes. This conversion strategy of WRN to CQDs/TiO_2_ [44] or CQDs/ZnO [45] composites added significant value and offered promising prospects for industrialization, potentially providing a new method for cooking waste recycling. However, although both catalysts could be recycled and reused multiple times without significantly reducing their photocatalytic degradation capacity, each recycling step requires careful centrifugation, washing, and drying. This process posed significant challenges when applied to large-scale wastewater treatment in practical settings.

On the other hand, iron oxide scale (IOS [46]) is formed on the surface of steel during the steel rolling process when the steel is rapidly cooled with water, resulting in iron-containing oxides. If not properly treated, significant amounts of IOS may be released into the environment, potentially causing severe pollution to water bodies, soil, and other natural surroundings [47]. At present, the primary method for handling waste IOS is to reintroduce it into the metallurgical industry, where it is used in the smelting of steel [48]. This well-established approach is suitable for large-scale production, but it requires significant investments in terms of funds, equipment, land, and labor. Furthermore, there is the possibility that waste IOS may contain Ca, Mg, P, and Si, providing an alternative pathway for reusing it as fertilizer [49]. However, such fertilizers can only utilize a small fraction of the effective components present in the IOS, leading to generally low fertilizer efficiency. Considering these limitations, it is essential to explore novel approaches that utilize waste IOS as a raw material to produce high-tech, high-value functional materials.

Based on previous research, our goal is to design and synthesize a new functional material derived from waste that boasts lower production costs and enhanced recyclability. We proposed integrating two types of waste materials, namely rice noodle waste (WRN) and iron oxide scale (IOS), as the raw materials. Firstly, the WRN underwent a hydrothermal process to convert it into CQDs, while the IOS was calcined to obtain iron oxides (FeO_x_). Subsequently, we combined the CQDs from WRN with the FeO_x_ from IOS, achieving the pioneering synthesis of a cost-effective magnetic CQDs/FeO_x_ photocatalytic composite material without the need for additional synthetic agents (see Figure 1). This material exhibited a good photocatalytic performance and could be effectively separated using magnets, showcasing its commendable recyclability and reusability. Our research delves into the material’s structure, photocatalytic performance, and photocatalytic mechanism, as well as thoroughly analyzing the effects of IOS calcination and CQDs combination on the material’s properties.

## 2. Experiment Section

### 2.1. Materials

The waste rice noodle (WRN) was obtained from the canteen located at the Guilin University of Technology in Guilin, China, with its primary organic components including starch (21.36 g/100 g), protein (1.91 g/100 g), and fat (0.4 g/100 g). The waste iron oxide scale (IOS, 200 mesh, main element content: Fe 71.09 wt%, O 27.41 wt%, Si 0.47 wt%, Ca 0.42 wt%, Mn 0.38 wt%, Al 0.23 wt%) was obtained from Valin Lianyuan Iron and Steel Co. Ltd. in Hunan, China. The tetracycline hydrochloride (98% purity), methylene blue (98.5% purity), nano ZnO (99% purity), nano TiO_2_ (99% purity), 1,4-benzoquinone (BQ, 98% purity), 2-propanol (IPA, 99% purity), ethylenediaminetetraacetic acid disodium salt (EDTA–2Na, 98% purity), as well as dimethyl pyridine N-oxide (DMPO, 99% purity), were procured from Macklin Reagent (Shanghai, China) and were used as received, without undergoing additional purification steps.

### 2.2. Synthesis

To initiate the process, the formulation of the CQDs solution commenced via a hydrothermal treatment applied to the WRN, adhering to the methodology outlined in our earlier research [44,45]. In the standard synthesis protocol, 100 g of WRN underwent thorough grinding to yield a consistent paste using a mortar. Subsequently, this paste was combined with 200 g of deionized water. The ensuing mixture underwent heating within a 500 mL Teflon-lined autoclave, where it was subjected to a temperature of 200 °C for a duration of 10 h. A brown CQDs solution, accompanied by black-gray sediment, was successfully acquired. Employing vacuum filtration, the brown CQDs solution was isolated, intended for subsequent fabrication of CQDs/FeO_x_ photocatalytic composites, while the solid portion was separated to yield hydrothermal carbon (HTC) powder, which could be further treated to produce activated carbon [44]. Simultaneously, the IOS powder underwent calcination at various temperatures (100 °C, 200 °C, 300 °C, 400 °C, 500 °C) in an air atmosphere using a SGM6812CK tube furnace (Sigma, Luoyang, China) for 4 h, producing FeO_x_ as a brown powder.

Next, a volume of 40 mL from the CQDs solution was skillfully combined with an appropriate quantity of FeO_x_, choosing from three different amounts (0.25 g, 0.5 g, 1 g), in precisely measured proportions. Through meticulous magnetic stirring at room temperature for a duration of 0.5 h, a consistent and homogenous suspension was created. Subsequently, this well-mixed composition was placed into a 50 mL Teflon-lined autoclave, undergoing heating at a temperature of 85 °C for a span of 3 h to facilitate the formation of the composite structure. Following this process, the composite material was retrieved and subjected to centrifugation, followed by thorough triple washing with distilled water. Upon being subjected to vacuum drying at 60 °C, the resulting product emerged as the CQDs/FeO_x_ photocatalytic composite material, taking the form of a deep brown powder. The samples were labeled as CQDs/FeO_x_–1 to CQDs/FeO_x_–7 based on the calcination temperature of the IOS and the amount of FeO_x_ used. The relevant formulation design and elemental composition are presented in Table 1.

### 2.3. General Characterization and Measurement of Photocatalytic Performance

The characterizations and photocatalytic degradation experiments of the CQDs/FeO_x_ composite material were conducted following the methods outlined in our previous reports [44,45]. For in-depth details, please refer to Appendix A in the ESI.

## 3. Result and Discussion

### 3.1. Structural Characterization

The PXRD analysis of the CQDs/FeO_x_ composite (CQDs/FeO_x_–3 sample), FeO_x_ powder (achieved through the calcination of IOS at 300 °C), and IOS (raw material) is depicted in Figure 2. Evidently, the waste IOS contained two main phases. Among them, the characteristic peaks at 30.0°, 35.4°, 37.0°, 43.1°, 56.9°, and 62.5° corresponded to the (220), (311), (222), (400), (511), and (440) crystal planes, respectively, which could be attributed to magnetite (Fe_3_O_4_, JCPDS card no. 19-0629). Additionally, the characteristic peaks at 33.1° and 35.6° could be attributed to the (104) and (110) crystal planes of hematite (α-Fe_2_O_3_, JCPDS card no. 33-0664). This indicated that the IOS mainly contained Fe_3_O_4_ and α-Fe_2_O_3_, with Fe_3_O_4_ being dominant. Combining this with the results of the elemental composition analysis (See Table 2), the Fe/O mass ratio in the IOS was 2.59, and it could be calculated that there was only about 10.8% of α-Fe_2_O_3_ in the mixed phase of iron oxide. Typically, the band gap of magnetite (Fe_3_O_4_) was too small to exhibit an effective photocatalytic ability. As a result, the iron oxide photocatalytic component was mainly α-Fe_2_O_3_. Therefore, it was foreseeable that the waste IOS could not be directly used as a photocatalytic material or directly combined with CQDs due to the low content of α-Fe_2_O_3_, making calcination treatment necessary to enhance its photocatalytic potential.

Comparing the PXRD pattern of the FeO_x_ powders obtained through the calcination of IOS at 300 ℃ with that of IOS, it was evident that the diffraction peaks of the α-Fe_2_O_3_ were significantly enhanced after calcination. The characteristic peaks at 24.1°, 33.1°, 35.6°, 40.9°, 49.5°, and 54.1° corresponded to the (012), (104), (110), (113), (024), and (116) crystal planes of α-Fe_2_O_3_, respectively. Additionally, based on compositional analysis, the Fe/O mass ratio decreased to 2.51, indicating that the FeO_x_ powder contained 43.3% of α-Fe_2_O_3_. This transformation implied that calcination in air could convert a portion of the initially non-photocatalytic Fe_3_O_4_ into photocatalytically active α-Fe_2_O_3_, which laid the foundation for obtaining practical photocatalytic materials after the composite with CQDs. The PXRD patterns of the CQDs/FeO_x_ composite and FeO_x_ exhibited resemblances; however, the diffraction peaks corresponding to the CQDs within the composite were not prominently discernible due to their relatively modest concentration.

The results of the particle size distribution testing indicated that the obtained CQDs/FeO_x_ composite (CQDs/FeO_x_–3 sample) exhibited a Z-average particle size value of 127.8 ± 63.26 nm (refer to Appendix A in ESI). The outcomes of the TEM analysis for the CQDs/FeO_x_ composite (CQDs/FeO_x_–3 sample) are visually depicted in Figure 3a. It could be observed that the CQDs/FeO_x_ composites exhibited an irregular lamellar structure, with uniformly dispersed spherical CQDs particles on the FeO_x_ surface. In Figure 3b, the HRTEM image vividly revealed the intricate lattice arrangement of the FeO_x_ and CQDs. The lattice stripes, exhibiting spacings of 0.185 nm, 0.198 nm, and 0.271 nm, corresponded, respectively, to the (104), (110), and (024) crystal planes of α-Fe_2_O_3_. Furthermore, discernible lattice stripes with spacings of 0.103 nm and 0.168 nm aligned with the (111) and (220) crystal planes of Fe_3_O_4_. These observations indicated that the obtained FeO_x_ was primarily a mixture of magnetite (Fe_3_O_4_) and hematite (α-Fe_2_O_3_), which aligned with the PXRD results. Moreover, crystalline planes with a lattice spacing of about 0.283 nm, corresponding to the (020) crystalline planes of CQDs [50,51], were also observed, confirming the successful combination of the WRN-based CQDs and FeO_x_ in the resulting composite.

The XPS spectra of both the CQDs/FeO_x_ composite and FeO_x_ powders are displayed in Figure 4 and summarized in Table 3. Figure 4a highlights the elemental presence of Fe, O, and C within the CQDs/FeO_x_ composite. Notably, Figure 4b and Table 2 elucidate that the Fe 2p spectra disclosed characteristic peaks at 724.67 and 710.49 eV, corresponding to the Fe(2p_1/2_) and Fe(2p_3/2_) signals, respectively. A noteworthy separation of divalent and trivalent iron signals was discernible within the Fe(2p_3/2_) signal, indicative of the presence of both Fe^2+^ and Fe^3+^ ions in the resulting CQDs/FeO_x_ composite. In the high-resolution spectrum of O 1s, the CQDs/FeO_x_ complex displayed distinct Fe–O and C–O bonds, hosting characteristic signals positioned at 530.71 and 528.29 eV, respectively. In contrast, the O 1s spectrum of the FeO_x_ calcined at 300 °C displayed two distinct peaks at 530.5 and 529.38 eV, attributing them to the Fe–O bond and the hydroxyl group present on the FeO_x_ surface, respectively. This led to the inference that the interaction between CQDs and FeO_x_ involved the carboxyl group within the CQDs and the Fe–OH group on the surface of the FeO_x_, leading to the vanishing of the surface hydroxyl signal and the emergence of the C–O bond signal, as indicated in Figure 4c and Table 3. Additionally, in the high-resolution C 1s spectrum of the CQDs/FeO_x_ composite, the peak at 283.69 eV could be attributed to the C–C bond within the CQDs. Simultaneously, the signals centered at 285.03 and 287.51 eV corresponded to the C–O and C=C bonds of CQDs, respectively, confirming the successful integration of the CQDs and FeO_x_ within the resultant composites.

As shown in Figure 5a, without the combination of CQDs, the IOS and FeO_x_ exhibited an obvious Fe–OH stretching vibration peak at around 880 cm^−1^ in their IR spectra. However, this peak disappeared after the composite of FeO_x_ with CQDs. Paired with the XPS findings, it is conceivable that the intricate interplay between the CQDs and FeO_x_ constituted a response involving the carboxyl group within the CQDs and the hydroxyl group located on the FeO_x_ surface. Furthermore, characteristic absorption peaks of CQDs were observed in the IR spectra of the CQDs/FeO_x_ composites, including the stretching vibration (3422 cm^−1^) and bending vibration (1620 cm^−1^) of the O–H bond on the CQDs, and the stretching vibration (1063 cm^−1^) of the C–O bond, etc. In addition, due to the presence of organic groups such as hydroxyl and carboxyl groups in the CQDs, the PZC value of the CQDs/FeO_x_ composites (8.46) was higher than that of the non-composite FeO_x_ powder (7.98) (See Figure 5b). In addition, due to larger feedstock particles, the CQDs/FeO_x_ composite exhibited a relatively low specific surface area (2.28 m^2^ g^−1^) compared with the CQDs/TiO_2_ [44] and CQDs/ZnO [45] photocatalytic composites in our previous report; however, it was still slightly higher than that of the non-composite FeO_x_ powder (1.57 m^2^ g^−1^) (See Figure 5c).

### 3.2. Photocatalytic Performance of CQDs/FeO_x_ Composites

The resulting CQDs/FeO_x_ composite exhibited excellent photocatalytic degradation efficiency towards various organic pollutants. Figure 6a underscores the formidable challenge of degrading methylene blue, a prevalent and highly toxic pollutant in dyeing wastewater, using 405 nm visible purple light. In the absence of a catalyst, its degradation rate remained at a mere 2.3%, even after an extended duration of 8 h (480 min). Due to its non-photocatalytic Fe_3_O_4_ component, the IOS itself lacked the ability for photocatalytic degradation to methylene blue, as its degradation rate was only 4.9% after 480 min of light irradiation, which was almost indistinguishable from the degradation without the catalyst. However, after calcination at 300 °C, a portion of the Fe_3_O_4_ in the IOS was converted into α-Fe_2_O_3_, resulting in the FeO_x_ with a certain photocatalytic degradation ability. The degradation rate of the FeO_x_ towards methylene blue reached 41.8% after 480 min of light irradiation, with an apparent degradation rate constant of 1.03 × 10^−3^ min^−1^. However, complete degradation could not be achieved under these conditions. In contrast, the catalytic effect of the CQDs/FeO_x_ composite (CQDs/FeO_x_–3 sample) on methylene blue significantly improved after incorporating the WRN-based CQDs into the FeO_x_. The composite demonstrated a good photocatalytic degradation rate (up to 99.30% within 480 min) and a relatively high degradation rate constant (5.26 × 10^−3^ min^−1^), enabling the complete degradation of methylene blue (See Appendix A in ESI).

It was found that achieving a good photocatalytic performance for the CQDs/FeO_x_ composite required the appropriate calcination of the IOS in an air atmosphere to enhance the content of α-Fe_2_O_3_. The degradation rates of methylene blue for CQDs/FeO_x_ composite materials obtained at different calcination temperatures (samples CQDs/FeO_x_–1 to CQDs/FeO_x_–5) are shown in Figure 6b,c. Combined with the results of the phase and elemental composition, it was observed that without proper high-temperature calcination to increase the content of α-Fe_2_O_3_, even with the incorporation of CQDs, the catalytic effect could hardly be improved. For instance, the CQDs/FeO_x_–1 sample obtained by heating the IOS at 100 °C had a Fe/O mass ratio of 2.59 due to the relatively low calcination temperature. It was found that only about 10.9% of α-Fe_2_O_3_ was present in the mixed FeO_x_ phase, similar to the IOS as a raw material. Consequently, despite containing CQDs, the photocatalytic degradation of CQDs/FeO_x_–1 remained poor. After 480 min of light exposure, its degradation rate to methylene blue was only 35.79%, and the degradation rate constant was merely 7.04 × 10^−4^ min^−1^, which is even worse than the FeO_x_ sample without CQDs but calcined at 300 °C. There are reports indicating that, when heated in air at an appropriate temperature (around 300 °C), Fe_3_O_4_ exhibits a tendency to transform into α-Fe_2_O_3_ [52,53,54]. As shown in Table 1, with enhancing the calcination temperature, the Fe/O mass ratio of the composite FeO_x_ material showed a trend of initially decreasing and then increasing, indicating that the content of α-Fe_2_O_3_ first increased and then decreased. Among them, the CQDs/FeO_x_–3 sample prepared with IOS calcined at 300 °C had the smallest Fe/O mass ratio (2.51), which corresponded to the highest α-Fe_2_O_3_ content (44%). Therefore, it exhibited the best photocatalytic performance and achieved the complete degradation of methylene blue within 480 min, with a degradation rate constant of 5.26 × 10^−3^ min^−1^. Furthermore, the material demonstrated excellent recyclability, and its degradation rate remained above 98% even after ten cycles of photocatalysis (See Figure 6d). However, when compared to commercially available nanoscale TiO_2_ or ZnO, the photocatalytic degradation efficiency of the obtained CQDs/FeO_x_ photocatalytic material was relatively low (See Figure 6e,f). Nevertheless, due to its complete reliance on waste materials as feedstock, the cost of the CQDs/FeO_x_ photocatalytic material is significantly lower. Additionally, it possesses strong magnetism, unlike commercial TiO_2_ or ZnO, enabling convenient recovery. Therefore, it may present a more competitive option for large-scale water purification.

Additionally, the loading amount of CQDs also affects the photocatalytic degradation capability of the CQDs/FeO_x_ composite material. As shown in Figure 6g,h, at lower carbon contents, the photocatalytic degradation efficiency of the CQDs/FeO_x_ composite material improved with the increasing carbon content. The CQDs/FeO_x_ composite material with a carbon content of approximately 11 wt% (CQDs/FeO_x_–3 sample) exhibited the best photocatalytic performance, with a degradation rate constant of 5.26 × 10^−3^ min^−1^ for methylene blue. However, it was found that further increasing the loading amount of CQDs may have an adverse effect on the photocatalytic performance. For example, in the case of the CQDs/FeO_x_–7 sample featuring a carbon content of 20.2 wt%, a comparatively reduced degradation rate constant of 3.73 × 10^−3^ min^−1^ was observed for methylene blue degradation. This phenomenon could be attributed to the potential shielding influence stemming from the presence of carbon-based constituents [55].

The obtained CQDs/FeO_x_ composite material could also be utilized for controlling antibiotic residues, as shown in Figure 7. Under 405 nm purple light, the CQDs/FeO_x_ composite (CQDs/FeO_x_–3 sample) could degrade 98.21% of tetracycline within 320 min, with an apparent degradation rate constant of 3.73 × 10^−3^ min^−1^. In comparison, under the same conditions, the FeO_x_ powder without CQDs could only degrade 40.45% of tetracycline, with an apparent degradation rate constant of 1.29 × 10^−3^ min^−1^.

Compared with other CQDs/iron oxide composites using different iron and carbon sources (Table 4), the CQDs/FeO_x_ composite reported in this paper demonstrated a comparably good photocatalytic performance. Moreover, due to the complete utilization of waste materials (IOS and WRN) as raw sources, the synthesis cost of this material was significantly reduced compared to other CQDs/iron oxide composites [29,30,31,32,33,34]. This complete conversion of waste to treasure makes it more environmentally friendly and exhibits outstanding sustainability, making it more suitable for large-scale, industrial applications in water purification projects. In comparison to our previous reports on CQDs/TiO_2_ and CQDs/ZnO composites based on WRN, the CQDs/FeO_x_ composite exhibited a weaker photocatalytic performance (See Table 5). However, due to the presence of magnetic Fe_3_O_4_ components, it offers the advantage of simplified recovery procedures while ensuring high recyclability. Additionally, as the oxide component is derived from waste IOS, it eliminates the need for commercial reagents, further reducing the costs and promoting environmental friendliness.

### 3.3. Magnetic Properties of CQDs/FeO_x_ Composites

In addition to exhibiting efficient photocatalytic degradation to various organic pollutants, the resulting CQDs/FeO_x_ composite also demonstrated outstanding magnetic properties. The magnetization curves of the CQDs/FeO_x_ composite and the raw IOS were tested and are shown in Figure 8a. The IOS powder exhibited remarkable soft ferromagnetism at room temperature, characterized by a low coercivity of 97.4 Oe, a low residual magnetization intensity of 5.2 emu/g, and a high saturation magnetization intensity of 5.8 emu/g, primarily attributed to the abundance of Fe_3_O_4_ particles in the material.

As mentioned earlier, after the process of calcination and compounding, the magnetic properties of the CQDs/FeO_x_ composites slightly weakened due to the reduced Fe_3_O_4_ content. Nevertheless, its saturation magnetization intensity remained at approximately 54.6 emu/g, along with low coercivity (117.1 Oe) and residual magnetization intensity (5.8 emu/g). Therefore, the CQDs/FeO_x_ composite still exhibits exceptional soft magnetic characteristics, making it a promising candidate for various magnetic applications.

As depicted in Figure 8b,c, the magnetic properties of the CQDs/FeO_x_ composite material enable efficient and convenient recyclability in photocatalytic applications. Following the photocatalytic process, the CQDs/FeO_x_ composite can be effortlessly separated using a magnetic field, achieving a mass recovery rate of 99.62% after the initial photocatalytic cycle and 98.45% after ten cycles of photocatalysis. Additionally, iron oxide photocatalysts have demonstrated excellent environmental compatibility, with minimal toxicity to fish and algae [56,57,58]. Consequently, the resulting CQDs/FeO_x_ photocatalyst holds significant promise for large-scale industrial water purification.

### 3.4. Photocatalytic Mechanism of CQDs/FeO_x_ Composites

To delve deeper into the photocatalytic mechanism of the resulting CQDs/FeO_x_ composite, the UV-VIS diffuse reflectance spectra of both the FeO_x_ and the CQDs/FeO_x_ composite are illustrated in Figure 9a. Evidently, even after calcination at 300 °C, the main energy absorption region of the FeO_x_ inorganic phase was in the ultraviolet region, with almost no absorption in the visible wavelength range above 380 nm. In contrast, the CQDs/FeO_x_ composite material exhibited strong absorption throughout the entire visible light region, indicating that the introduction of CQDs enabled the composite material to utilize the energy in the visible light region more effectively, thereby generating more electron-hole pairs. Moreover, the determination of band gaps for the FeO_x_ and CQDs/FeO_x_ samples was performed through the application of the Tauc plot method [59,60], as depicted in Figure 9b. The band gap of the FeO_x_ was 2.17 eV, but after introducing CQDs, the band gap of the CQDs/FeO_x_ composite material was further reduced to 1.40 eV. This narrower band gap allowed for more the effective utilization of energy in the visible light region, significantly promoting electron transitions and enhancing the photocatalytic degradation performance.

The valence band (VB) potentials of the CQDs/FeO_x_ composite and FeO_x_ powder are determined using XPS valence spectra. As shown in Figure 9c, due to the presence of Fe_2_O_3_, the VB potential of the FeO_x_ powder was 2.59 eV, which was more positive than E^0^(·OH, H^+^/H_2_O) (2.38 eV vs. NHE). This suggested that the FeO_x_ powder could oxidize water to generate hydroxyl radical (·OH). However, the conduction band (CB) potential (E_CB_ = E_VB_ − E_g_) of the FeO_x_ was determined to be 0.42 eV, which was more positive than the standard electrode potential E^0^(O_2_, H^+^/·O_2_H) for superoxide radicals (−0.046 eV vs. NHE). This indicated that the FeO_x_ powder could not reduce oxygen in water to produce superoxide radical (O_2_^·–^). On the other hand, the introduction of CQDs raised the VB potential of the CQDs/FeO_x_ composite material to 2.83 eV, enabling better oxidation of water to form photocatalytically active hydroxyl radical. Combining its own band gap (1.40 eV) and the band gap of CQDs (2.14 eV), the CB potential was calculated to be −0.08 eV, which was lower than E^0^(O_2_, H^+^/·O_2_H) for superoxide radicals (−0.046 eV vs. NHE). This indicated that the CQDs/FeO_x_ composite material could reduce oxygen in water to generate photocatalytically active superoxide radical.

The photoluminescence emission profiles of both the CQDs/FeO_x_ composite material and FeO_x_ are presented in Figure 9d. The fluorescence emission intensity exhibited by the CQDs/FeO_x_ composite material was notably subdued in comparison to that of the FeO_x_. This observation implies that the incorporation of CQDs could proficiently curtail the recombination of photogenerated electron-hole pairs, thereby significantly contributing to the enhancement of the photocatalytic degradation efficacy. The enhanced mechanism of the CQDs/FeO_x_ composite material on the catalytic performance of the FeO_x_ was also demonstrated through transient photocurrent response (PCR) under visible light irradiation and electrochemical impedance spectra (EIS). As shown in Figure 9e, under purple light illumination, the photocurrent intensity of the CQDs/FeO_x_ composite material was approximately 11 times that of the FeO_x_ without CQDs, indicating that the CQDs/FeO_x_ composite could achieve a more efficient interface charge transfer and more effective electron-hole pair separation. The lower probability of photogenerated electron-hole recombination resulted in a significant improvement in the photocatalytic degradation performance. Additionally, the Nyquist plot of the CQDs/FeO_x_ composite displayed a smaller semicircle diameter than that of the FeO_x_ powder, indicating that the resulting CQDs/FeO_x_ composite exhibited lower charge transfer resistance than the FeO_x_ powder. This ensured a more efficient interface charge transfer and more effective electron-hole pair separation, consistent with the photocurrent analysis results (see Figure 9f).

The impacts of distinct quenching agents (EDTA–2Na, BQ and IPA) on the photodegradation process of methylene blue are elucidated in Figure 10a,b. Notably, the introduction of EDTA–2Na marginally curtailed the photocatalytic degradation efficacy of the CQDs/FeO_x_ composite material, yielding a photocatalytic efficiency of 93.44% in comparison to the absence of quenching agents. This observation underscores that photogenerated holes (h+) played a minor role and were not the primary drivers of the photocatalytic activity. In contrast, the incorporation of BQ or IPA substantially impeded the degradation efficiency, yielding photocatalytic efficiencies of 40.75% and 60.15%, respectively, as opposed to the scenario without quenching agents. This phenomenon implies that both superoxide radicals (O_2_^·–^) and hydroxyl radicals (·OH) stood as the predominant active species in the photocatalytic degradation mechanism, with O_2_^·–^ playing a more pronounced role. Furthermore, using DMPO as a radical trapping agent, electron spin resonance spectroscopy (ESR) was carried out to study the active oxygen species generated by the CQDs/FeO_x_ composite and FeO_x_. As shown in Figure 10c–f, the addition of the CQDs/FeO_x_ composite resulted in strong characteristic peaks of both superoxide radical (O_2_^·–^) and hydroxyl radical (·OH). This indicates that the CQDs/FeO_x_ composite material could reduce adsorbed O_2_ to form superoxide radical (O_2_^·–^) and oxidize adsorbed H_2_O to form hydroxyl radical (·OH) under light irradiation. In contrast, under the same test conditions, the FeO_x_ powder could not generate superoxide radical (O_2_^·–^) effectively, while its signal of hydroxyl radicals was weak.

As shown in Figure 10g, the possible photocatalytic mechanism of the CQDs/FeO_x_ composite material was similar to that of the CQDs/TiO_2_ [44] and CQDs/ZnO [45] composites in our previous report. Upon exposure to visible light, the CQDs/FeO_x_ composite undergoes a dynamic process: the CQDs become readily excited by photogenerated electrons situated in the conduction band (CB), leaving behind holes in the valence band (VB). This excitation triggers rapid spatial electron transfer between the CQDs and FeO_x_ particles, effectively suppressing recombination and yielding the enhanced separation of electron-hole pairs. As a result, photogenerated electrons amass in the CB of the CQDs, while the holes populate the VB of FeO_x_, with each entity primed for their distinct roles in photocatalytic reactions. Photogenerated holes engage with H_2_O to yield a profusion of ·OH radicals, while photogenerated electrons react with O_2_, leading to an abundant production of O_2_^·–^ radicals. These generated O_2_^·–^ and ·OH radicals collectively orchestrate the degradation of diverse organic pollutants, thereby showcasing exceptional prowess in the realm of photocatalytic degradation activity.

## 4. Conclusions

In this study, we successfully synthesized low-cost CQDs/FeO_x_ composites by combining waste rice noodle (WRN) and iron oxide scale (IOS) and assessed their photocatalytic performance. The key steps in our synthesis strategy were identified: first, calcining IOS in an air atmosphere to enhance the photocatalytically active α-Fe_2_O_3_ content; second, incorporating WRN-based CQDs into FeO_x_ to enhance the electron-hole pair separation, leading to increased O_2_ reduction and H_2_O oxidation. This significantly improved the photocatalytic performance. The resulting CQDs/FeO_x_ composites efficiently degraded various organic pollutants under purple light irradiation and had superior magnetic properties, allowing for easy separation and reuse.

Our CQDs/FeO_x_ composite achieved the 100% conversion of WRN and IOS waste into high-value functional materials without additional synthetic additives. Compared to other reported photocatalytic composites, such as the CQDs/TiO_2_ and CQDs/ZnO from our previous work and other CQDs/iron oxide composites using synthetic reagents, our CQDs/FeO_x_ composite is more cost-effective, environmentally friendly, and sustainable. Its magnetic properties enable rapid separation and efficient reuse, making it a promising candidate for large-scale, low-cost photocatalytic water purification with commercial potential. Furthermore, our approach to transform WRN and IOS into a CQDs/FeO_x_ composite opens new possibilities for the combined utilization of waste.

## Figures and Tables

**Figure 1 nanomaterials-13-02506-f001:**
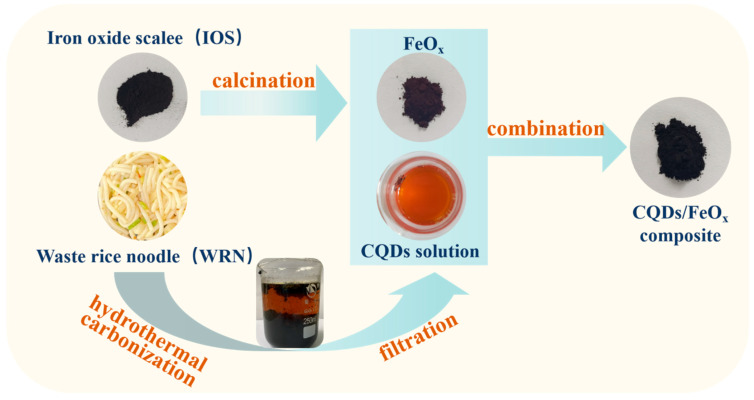
Formation process of CQDs/FeO_x_ composite using WRN and IOS as raw materials.

**Figure 2 nanomaterials-13-02506-f002:**
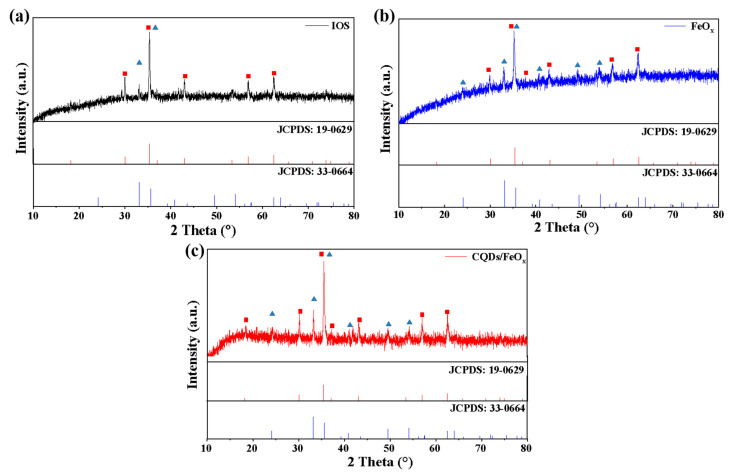
PXRD pattern of IOS (**a**), FeO_x_ (**b**) and CQDs/FeO_x_ composite (**c**). The red square and blue triangle represent the diffraction peaks of Fe_3_O_4_ and α-Fe_2_O_3_, respectively.

**Figure 3 nanomaterials-13-02506-f003:**
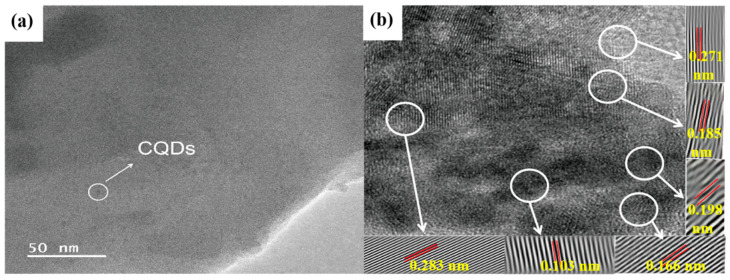
The TEM image (**a**) and HRTEM image (**b**) of CQDs/FeO_x_ composite based on WRN and IOS.

**Figure 4 nanomaterials-13-02506-f004:**
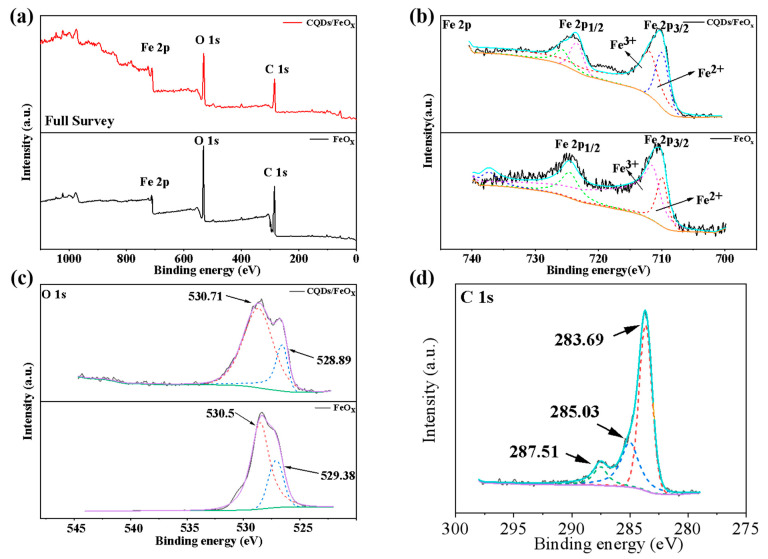
The full XPS (**a**), as well as Fe 2p ((**b**), black: original data, blue: overall fitting curve, green: Fe^3+^ curve at Fe 2p_1/2_, purple: Fe^2+^ curve at Fe 2p_1/2_, red: Fe^3+^ curve at Fe 2p_3/2_, navy blue: Fe^2+^ curve at Fe 2p_3/2_, orange curve: bottom line), O 1s ((**c**), black: original data, purple: overall fitting curve, red and blue: peak fitting, green: bottom line) and C 1s ((**d**), CQDs/FeO_x_ composite only, black: original data, blue: overall fitting curve, red and green: peak fitting, purple: bottom line) high-resolution spectrum of FeO_x_ powder and CQDs/FeO_x_ composite.

**Figure 5 nanomaterials-13-02506-f005:**
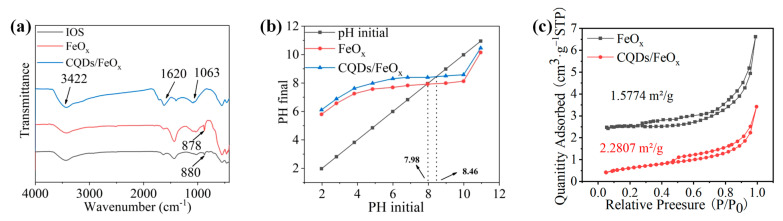
(**a**) IR spectra of IOS, FeO_x_ powder and CQDs/FeO_x_ composite; (**b**,**c**) The PZC value (**b**) and BET (**c**) of FeO_x_ powder and CQDs/FeO_x_ composite.

**Figure 6 nanomaterials-13-02506-f006:**
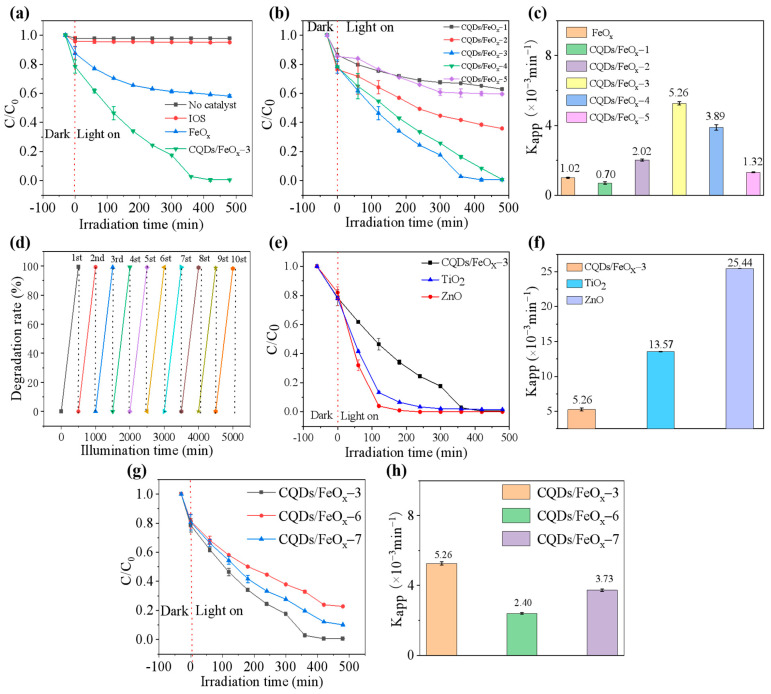
(**a**) Photocatalytic degradation rates of IOS, FeO_x_ powder and CQDs/FeO_x_ composites (CQDs/FeO_x_–3 sample) at 405 nm purple light for methylene blue at different irradiation times; (**b**) Photocatalytic degradation rates of CQDs/FeO_x_ composites prepared at different calcination temperatures for methylene blue at 405 nm purple light at different irradiation times; (**c**) Apparent degradation rate constants (*K_app_*) of CQDs/FeO_x_ composites and FeO_x_ powder prepared at different calcination temperatures; (**d**) Photocatalytic degradation performance of CQDs/FeO_x_ composites across varying photocatalytic cycles employing a 500-min operational cycle; (**e**,**f**) The photocatalytic degradation rates (**e**) and apparent degradation rate constant (*K_app_*, (**f**)) of CQDs/FeO_x_–3 sample, commercial TiO_2_ and ZnO; (**g**,**h**) Photocatalytic degradation rates (**g**) and apparent degradation rate constant (*K_app_*, (**h**)) of CQDs/FeO_x_ composites with different carbon contents.

**Figure 7 nanomaterials-13-02506-f007:**
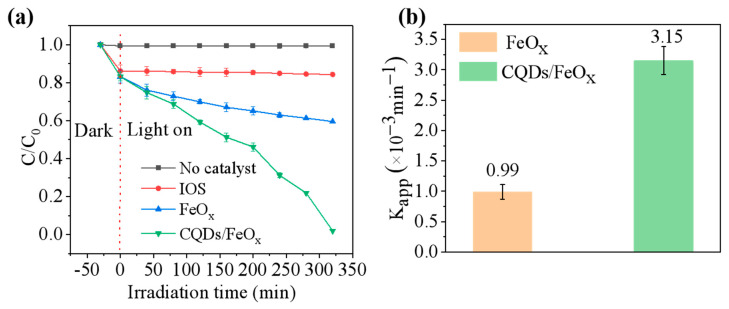
(**a**) Photocatalytic degradation rates of IOS, FeOx powder and CQDs/FeOx composite (CQDs/FeO_x_–3 sample) at 405 nm purple light for tetracycline hydrochloride at different irradiation times; (**b**) Apparent degradation rate constants (*K_app_*) of CQDs/FeOx composite (CQDs/FeO_x_–3 sample) and FeO_x_ powder.

**Figure 8 nanomaterials-13-02506-f008:**
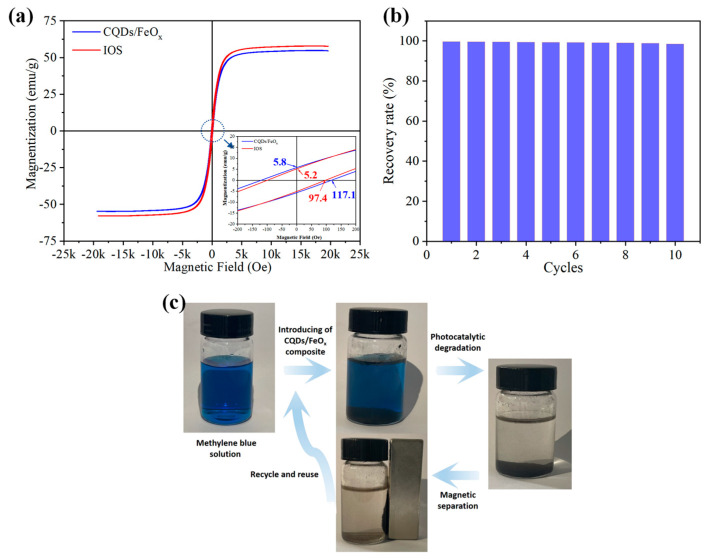
(**a**) Magnetization curves of IOS and CQDs/FeO_x_ composite (CQDs/FeO_x_–3 sample); (**b**) Mass recovery rate of CQDs/FeO_x_ composite (CQDs/FeO_x_–3 sample) across varying photocatalytic cycles; (**c**) Schematic illustration of the recycle and reuse of CQDs/FeO_x_ composite using magnetic separation.

**Figure 9 nanomaterials-13-02506-f009:**
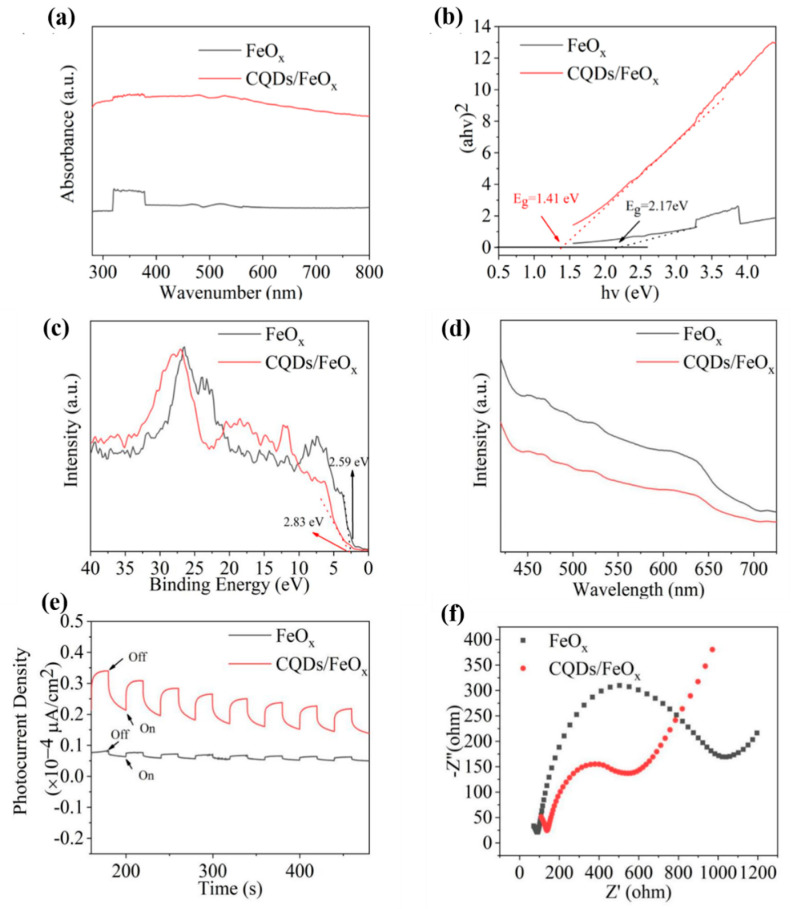
The UV-vis absorption spectra (**a**), Tauc plot curves (**b**), VB XPS spectra (**c**), PL spectra (**d**), PCR (**e**) and EIS (**f**) of the CQDs/FeO_x_ composite and FeO_x_.

**Figure 10 nanomaterials-13-02506-f010:**
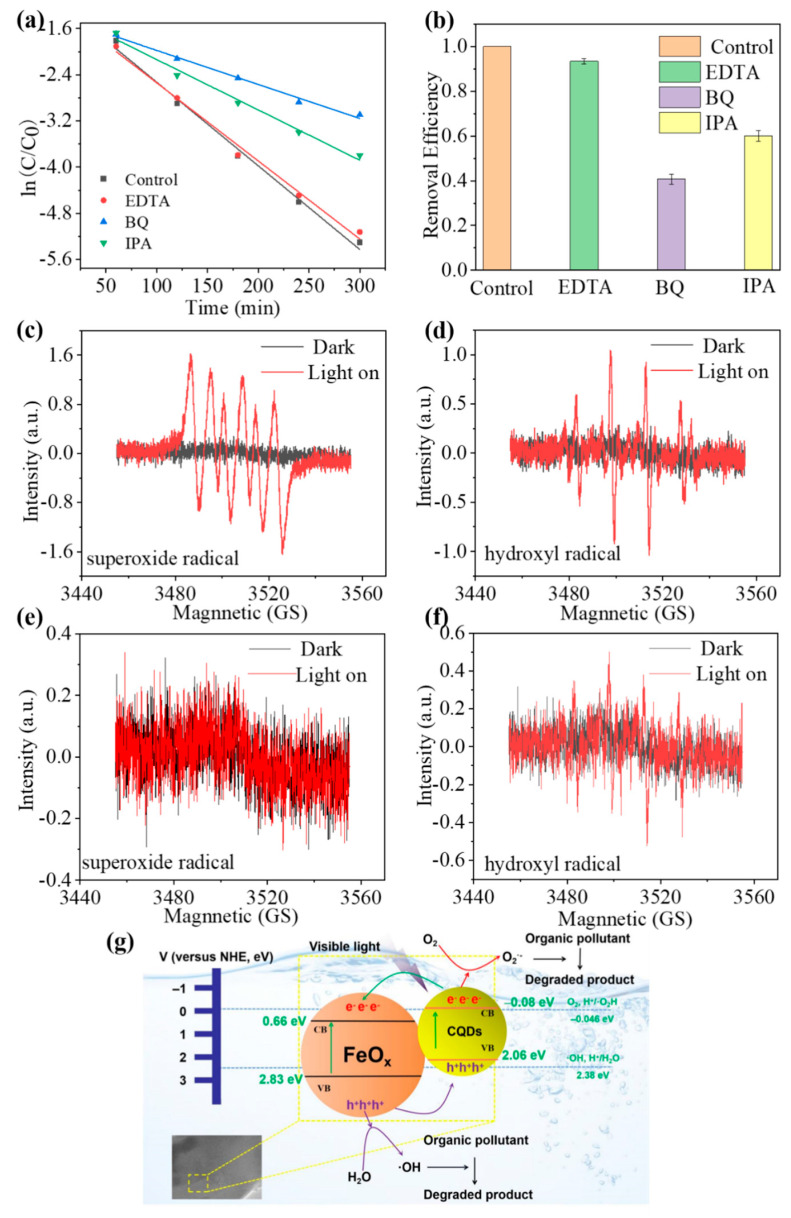
(**a**,**b**) Impact of various quenching agents (EDTA–2Na, BQ or IBA) on the photodegradation of methylene blue under 405 nm purple light. (**c**,**d**) ESR spectra of the CQDs/FeO_x_ composite in methanol (**c**) and water (**d**) using DMPO as a radical trapping agent. (**e**,**f**) ESR spectra of FeO_x_ in methanol (**e**) and water (**f**) using DMPO as a radical trapping agent. (**g**) Diagram illustrating the photocatalytic mechanism of the CQDs/FeO_x_ composite.

**Table 1 nanomaterials-13-02506-t001:** Formulation design of CQDs/FeO_x_ composites.

Serial Number	Calcination Temperature of IOS (℃)	Dosage of FeO_x_ (g)	Elemental Composition (wt%)						
			Fe	O	C	Mn	Al	Si	Ca
CQDs/FeO_x_–1	100	0.5	63.17	24.36	11.23	0.19	0.41	0.35	0.29
CQDs/FeO_x_–2	200	0.5	62.75	24.66	11.34	0.2	0.42	0.36	0.27
CQDs/FeO_x_–3	300	0.5	62.6	24.94	11.14	0.21	0.44	0.35	0.32
CQDs/FeO_x_–4	400	0.5	62.91	24.87	10.89	0.23	0.44	0.34	0.32
CQDs/FeO_x_–5	500	0.5	62.95	24.58	11.24	0.22	0.39	0.33	0.29
CQDs/FeO_x_–6	300	0.25	56.24	22.38	20.21	0.19	0.4	0.32	0.26
CQDs/FeO_x_–7	300	1	65.88	26.2	6.56	0.22	0.46	0.35	0.33

**Table 2 nanomaterials-13-02506-t002:** Comparison of composition element content of CQDs/FeO_x_ composite, FeO_x_ and IOS.

Serial Number	Elemental Composition (wt%)						
	Fe	O	C	Al	Si	Ca	Mn
IOS	71.09	27.41	-	0.23	0.47	0.42	0.38
FeO_x_ *^a^*	70.46	28.05	-	0.21	0.49	0.4	0.39
CQDs/FeO_x_ composite *^b^*	62.6	24.94	11.14	0.21	0.44	0.35	0.32

*^a^* calcining IOS at 300 °C in an air atmosphere for 4 h; *^b^* CQDs/FeO_x_–3 sample.

**Table 3 nanomaterials-13-02506-t003:** XPS peak distribution of FeO_x_ powder and CQDs/ FeO_x_ composite based on waste rice noodle.

Photocatalyst	Element	Peak (eV)	Surface Group	Assignment
CQDs/FeO_x_ composite *^a^*	C 1s	283.69	C	Graphitic carbon
		285.03	C–O	Alcoholic or etheric structure in CQDs
		287.51	C=C	Aromatic ring of CQDs
	O 1s	530.71	Fe–O	Oxygen bonded to iron
		528.29	C–O	Oxygen singly bonded to CQDs
	Fe 2p	710.49	Fe	Fe (2p_3/2_)
		724.67	Fe	Fe (2p_1/2_)
FeO_x_ *^b^*	O 1s	530.5	Fe–O	Oxygen bonded to iron
		529.38	Fe–OH	Surface hydroxyl group of FeO_x_
	Fe 2p	710.35	Fe	Fe (2p_3/2_)
		724.71	Fe	Fe (2p_1/2_)

*^a^* CQDs/FeO_x_–3 sample; *^b^* calcining IOS at 300 °C in an air atmosphere for 4 h.

**Table 4 nanomaterials-13-02506-t004:** Photocatalytic degradation performance of CQDs/iron oxide composites utilizing various iron and carbon sources.

Iron Source	Carbon Source of CQDs	Light Source	Pollutant	Pollutant Concentration	Photocatalyst Dosage (g/L)	Irradiation Times (min)	Degradation Rate (%)	Reference
Waste IOS	WRN	Purple light lamp (20 W, wavelength: 405 nm)	methylene blue	20 mg/L	2	480	99.30	This work
			tetracycline	20 mg/L	2	320	98.21	
Commercially-available magnietic Fe_3_O_4_ nanoparticles	Glucose	Xe lamp (400 W, wavelength > 420 nm)	methylene blue (in NaOH solution)	1 × 10^–3^ mol/L	1	30	83	[29]
Commercial γ-Fe_2_O_3_	Glucose	Xe lamp (300 W, wavelength: 455 nm)	sulfamethoxazole (SMX)	10 mg/L	0.2	120	95	[30]
FeSO_4_·7H_2_O	Citric acid	Xe lamp with a 420 nm cutoff filter (350 W)	tetracycline, (0.50 mM of H_2_O_2_ was added)	20 mg/L	0.25	60	93	[31]
Fe(NO_3_)_3_·9H_2_O	Citric acid	HPMVL visible light lamp (250 W)	Oxytetracycline	10 mg/L	0.2	100	98	[32]
FeSO_4_·7H_2_O	Citric acid	Xe lamp (300 W, wavelength > 420 nm)	Metronidazole	30 mg/L	0.2	45	99.36	[33]
FeCl_3_·6H_2_O	Citric acid and Ethylenediamine	Xe lamp (300 W, wavelength: 420 nm)	Methylene blue	20 mg/L	0.5	120	99.3	[34]

**Table 5 nanomaterials-13-02506-t005:** Photocatalytic degradation performance of different CQDs/metal oxide composites based on WRN.

Metal Source	Carbon Source of CQDs	Light Source	Pollutant	Pollutant Concentration	Photocatalyst Dosage (g/L)	Irradiation Times (min)	Degradation Rate (%)	Reference
Waste IOS	WRN	Purple light lamp (20 W, wavelength: 405 nm)	methylene blue	20 mg/L	2	480	99.30	This work
			tetracycline	20 mg/L	2	320	98.21	
Commercial TiO_2_	WRN	Purple light lamp (20 W, wavelength: 405 nm)	methylene blue	20 mg/L	4	80	99.87	[44]
Commercial ZnO	WRN	Purple light lamp (20 W, wavelength: 405 nm)	methylene blue	20 mg/L	2	10	98.88	[45]
			tetracycline	20 mg/L	2	10	98.21	

## Data Availability

Not applicable.

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
