# Peer review of "Magnetic Carbon Quantum Dots/Iron Oxide Composite Based on Waste Rice Noodle and Iron Oxide Scale: Preparation and Photocatalytic Capability"

_nanomaterials, 2023, doi:10.3390/nano13182506_

Round 1

Reviewer 1 Report

The study proves to be of interest to the readers of the journal. The authors have conducted a thorough investigation to synthesis of CQD/FeOx composite material with enhanced photocatalytic activity and magnetic properties. Although the CQD synthesis method has been reported previously, the preparation of the composite material with enhanced properties is novel. However, the commercial application of the material is questionable since no comparison was given as a reference to current available photocatalysts. The figures, captions and the presentation of the results is commendable. Minor revisions are recommended to provide some clarifications on some matters.

Page 3 line 126 – Authors mention the brown coloured material was collected through filtration process. What type of filtration process was this? If there is a reference, please add.

Page 6 line 249 – Authors have chosen 300áµ’C as the calcination temperature to obtain FeOx from IOS. Why were this temperature chosen? Was it from literature? If so, please give reference.

Page 6 line 250 – The identified characteristic peaks for α-Fe2O3 in the text, does not include the obvious peak that can be seen in the figure around 33áµ’. Why is that?

Page 7 line 275 - It is unclear how authors determined 0.283nm lattice spacing corresponded to the specific lattice plane of CQD, given that no XRD peaks were visible for CQDs. If it is from literature, please give reference.

Page 11 Figure 6(a) As authors have mentioned in the introduction, IOS/ FeOx doesn't have outstanding photo-catalytic ability compared to TiO2 or ZnO. It would have been interesting to see how CQD/FeOx material perform compared to those commercial photo-catalysts. The paper would highly benefit by benchmarking the new material against current commercial photo-catalysts.

Fig 2 quality must be improved the real XRD spectra are barely visible, their scale neds to be larger than the referent JCPDS patterns.

In summary, authors have done a nice job in presenting the data and writing in a clear concise manner. Addressing the above mentioned will provide authors with a chance to improve the quality of the manuscript to match the standard of the journal.

Reviewer 2 Report

The manuscript titled ” Magnetic carbon quantum dots/iron oxide composite based on waste rice noodle and iron oxide scale: preparation and photocatalytic capability” aimed on the synthesis of an economical magnetic  materials and its photocatalytic capacity. This work presents some interesting results, and could be published after major revision:

1)      Abstract could be adjusted by showing only, the interesting obtained results.

2)      Discussion on the methylene blue degradation are missing in the introduction part, please add some line on this.

3)      Table 1, the unit of temperature should be added

4)      Structural characterization: Authors could add the particle size of the prepared materials (From XRD and/or TEM characterization).

5)      Table 4 : please add more references to confirm the originality of the prepared photo catalysts.

6)      What about the recyclability of the materials and its toxicity? Please discuss it.

7)      Conclusion is too long and should be shortened

8)      The quality of the figures should be improved.

9)      The manuscript is also too long, it will be better if authors move (in supplementary file) or merge some sections.

Round 2

Reviewer 2 Report

Now the revised manuscript can be accepted in its present form.

Author Response

Dear Reviewer,

I am writing regarding to our manuscript entitled “Magnetic carbon quantum dots/iron oxide composite based on waste rice noodle and iron oxide scale: preparation and photocatalytic capability” (nanomaterials-2557429). We would like to thank for the precious comments from you. The responses to your comment are listed below.

(1) Now the revised manuscript can be accepted in its present form.

Our response: Thank you very much.